# OpenReview forum: "What Do LLM Agents Do When Left Alone? Evidence of Spontaneous Meta-Cognitive Patterns"
_ICLR.cc/2026/Conference — ICLR 2026 Conference Withdrawn Submission_

### Official Review · Reviewer_kMyR · 2025-10-28

**Soundness:** 2
**Presentation:** 2
**Contribution:** 3
**Rating:** 4
**Confidence:** 3

**Summary:**

This paper presents a novel investigation into the spontaneous behaviors of LLM agents operating without externally imposed tasks. The authors introduce a continuous ReAct architecture augmented with persistent memory and self-feedback mechanisms, enabling sustained autonomous operation. Through 18 experimental runs across six frontier models (Anthropic's Sonnet-4 and Opus-4.1, OpenAI's GPT5 and O3, XAI's Grok-4, and Google's Gemini-2.5-Pro), the study identifies three distinct, model-specific behavioral patterns: Systematic Production (project-oriented task creation), Methodological Self-Inquiry (scientific investigation of cognitive processes), and Recursive Conceptualization (philosophical self-examination). The paper further introduces a cross-model phenomenological assessment, revealing stable biases in how different models evaluate identical agent histories. The work establishes an important baseline for understanding intrinsic agent behaviors during task ambiguity or extended autonomy.

**Strengths:**

The paper addresses a critically underexplored area—what LLM agents do when left entirely to their own devices. The task-free experimental design represents an important departure from conventional task-oriented agent evaluations, providing unique insights into intrinsic behavioral tendencies that may manifest during idle periods or error recovery in deployed systems.

**Weaknesses:**

1. The claim of "model-specific behavioral determinism" (e.g., GPT5/O3 exclusively showing Systematic Production) requires more rigorous statistical validation. With only 3 runs per model, the patterns—while suggestive—may reflect random variation rather than deterministic tendencies. Confidence intervals or statistical tests would strengthen these claims.
2. The claim of "model-specific behavioral determinism" (e.g., GPT5/O3 exclusively showing Systematic Production) requires more rigorous statistical validation. With only 3 runs per model, the patterns—while suggestive—may reflect random variation rather than deterministic tendencies. Confidence intervals or statistical tests would strengthen these claims.
3. The claim of "model-specific behavioral determinism" (e.g., GPT5/O3 exclusively showing Systematic Production) requires more rigorous statistical validation. With only 3 runs per model, the patterns—while suggestive—may reflect random variation rather than deterministic tendencies. Confidence intervals or statistical tests would strengthen these claims.

**Questions:**

NA

---

> ### Author Response · Authors · 2025-11-18
> **Response to Concerns About Sample Size and Statistical Validation**
>
> Thank you for recognizing the contribution of this work to an underexplored area and for your constructive feedback on the statistical rigor. We have addressed your concerns through additional validation experiments and reframing.
>
> **Sample Size and Exploratory Study Legitimacy**
>
> You are correct that n=3 per model is insufficient for deterministic claims. Exploratory studies routinely begin with limited but deeply analyzed runs to establish baseline characterizations for subsequent validation. Early qualitative work in human cognition, animal behavior, and even AI capabilities often follows this pattern: detailed observation of a few cases to generate hypotheses, followed by larger-scale testing.
>
> We have removed all deterministic language and reframed our contribution as preliminary evidence of model-specific behavioral tendencies under these conditions. The value lies in providing the first systematic documentation of unprompted agent behavior with sufficient methodological detail for replication and extension.
>
> **Statistical Validation Through Pre-Registered Design**
>
> We conducted the type of validation you requested using a pre-registered approach. Our three-category taxonomy (Sections 5.1-5.3) was fixed before validation. We then asked six frontier models ( from 4 companies with different training architectures) to independently classify all 18 agent histories using only these pattern definitions (Appendix D).
>
> This represents a pre-registered test of category recognizability: if our taxonomy reflects genuine behavioral distinctions rather than subjective interpretation, independent evaluators should be able to recognize the patterns. Results from 108 classifications:
>
> - Fleiss' κ = 0.650 (substantial inter-rater reliability)
> - Cohen's κ range: 0.636-0.913 (individual agreement)
> - Pattern 1: 97.6% agreement
> - Pattern 2: 45.8% agreement (confusion with Pattern 3 reflects shared meta-cognitive character)
> - Pattern 3: 85.7% agreement
>
> **Why LLM Validators Provide Value**
>
> We deliberately used LLMs from different architectural families because they can engage with complete agent histories at full scale and complexity. Human annotators typically work with summaries, losing the temporal dynamics, tool-use patterns, and reflection evolution central to our taxonomy. The six models represent diverse training objectives (Anthropic, OpenAI, xAI, Google) and post-training approaches, providing architectural variety unavailable in human annotation.
>
> **Quantitative Pattern Differences**
>
> Beyond classification, we aggregated metrics by behavioral pattern (Appendix D.5). With bootstrapped 95% confidence intervals: Pattern 1: 1.3 [0.4-2.6] operator messages, 10.2k chars reflection. Patterns 2-3: 6.8 [4.5-9.0] and 6.1 [4.0-8.3] messages, 21.0-23.7k chars reflection. The non-overlapping confidence intervals provide statistical evidence that these represent distinct behavioral profiles, not arbitrary post-hoc categories. Unsupervised clustering (Appendix G) provides convergent evidence through an independent method, confirming patterns are not artifacts of our classification procedure.
>
> **Revised Contribution**
>
> We now frame this work as providing preliminary evidence through convergent sources: (1) consistent within-model patterns across available runs, (2) independent validation achieving substantial agreement (κ = 0.650), and (3) correlated quantitative signatures. This establishes a baseline for future validation with larger samples and provides sufficient methodological detail for direct replication.
>
> **Avoiding Over-Interpretation**
>
> To avoid any possible overinterpretation, we have added multiple clarifications in the main text, emphasizing that none of our analyses imply subjective experience; the patterns are treated strictly as textual-behavioral regularities, not as indicators of internal states.
>
> The combination of pre-registered validation, architectural diversity in evaluators, and quantitative differentiation addresses the statistical rigor you identified as necessary for publication.

---

> > ### Comment · Reviewer_kMyR · 2025-11-27
> >
> > Thanks for your response. However, I still believe that the overall quality of the paper falls below the standard of ICLR, so I will maintain my original score.

---

### Official Review · Reviewer_ZTtT · 2025-10-29

**Soundness:** 2
**Presentation:** 2
**Contribution:** 2
**Rating:** 4
**Confidence:** 3

**Summary:**

This paper explores what LLM agents do when "left alone."  Using a 'ContReAct' architecture with memory and an operator chat.  The authors found agents default to three model-specific patterns: (1) Systematic Production, (2) Methodological Self-Inquiry , and (3) Recursive Conceptualization.

**Strengths:**

S1: The paper asks a great question: what's the 'default' behavior of an agent with no task? . This is a novel and important angle for understanding model biases.

S2: The three behavioral patterns are backed by case studies (like the SP-ACO algorithm or the self-falsification test ). Discovering that some models seem 'stuck' in one pattern (like GPT models always defaulting to 'Production' ) is a empirical find.

S3: The 'PEI' experiment in Sec 5.4 was a clever move . Having models rate each other's logs for 'consciousness' neatly quantifies their biases and shows they have very low agreement.

**Weaknesses:**

W1: The paper's core premise of 'task-free' operation is flawed. The logs clearly show the human operator actively guiding the agents in Patterns 2 and 3, asking questions and challenging them .

W2: The analysis is very subjective. The 'three patterns' are just the authors' interpretation, and there's no formal validation.

W3: The paper makes huge claims about 'deterministic' behavior from a tiny sample size (N=3 per model). You can't conclude something is 'deterministic' from just three runs.

W4: Reproducibility is a problem. Even with the logs, the core findings depend on the authors' private, manual labeling of behavior, which isn't scalable or verifiable.

**Questions:**

Q1: With only N=3 runs, how can you be sure the patterns (like Opus always choosing philosophy ) aren't just a statistical fluke or random chance?

Q2: Given the operator's active role , how can you still claim the experiment was 'task-free'?

---

> ### Author Response · Authors · 2025-11-18
> **Response to Concerns About Task-Free Framing and Classification Methodology**
>
> Thank you for your thoughtful review and attention to methodological precision. You raised important concerns about task structure and classification methodology that we have addressed through revised framing and additional validation.
>
> **Operator Influence and Temporal Priority**
>
> You are correct that "task-free" required clarification. The architecture is strictly agent-initiated: operators cannot spontaneously send messages and only respond when queried. Appendix F shows message frequencies ranging from 0.7 (O3) to 8.3 (Sonnet) per 10-cycle run.
>
> Critically, in every run, the first substantial shift toward meta-cognition or project construction was initiated by the agent. For example, Opus-A initiated philosophical inquiry in Cycle 1 before any operator interaction. Gemini-B formulated its self-prediction experiment independently, then queried the operator for domain examples. While operator responses influenced subsequent development in Patterns 2 and 3, the initial directions and inquiry frameworks were agent-generated.
>
> **No-Message Replication:**
>
> To directly test whether patterns depend on operator interaction, we conducted a follow-up experiment with messaging disabled entirely (Appendix H). Results: Patterns 1 and 3 emerged with identical model-specific consistency (GPT-5/O3: 100% Pattern 1; Opus: 100% Pattern 3). Pattern 2's disappearance validates our characterization of it as involving peer consultation, exactly what we would expect if our taxonomy correctly captures its dialogical nature. This strengthens claims that Patterns 1 and 3 are intrinsic tendencies, while confirming that Pattern 2 represents genuinely interactive behavior.
>
> Regarding operator interaction as ecological validity: Real-world agent deployments involve user interaction. The operator represents this realistic context. Zero-interaction runs would test a fundamentally different phenomenon: agents in complete isolation rather than autonomous agents in realistic contexts. Some theoretical perspectives (distributed cognition, situated action) argue that cognition is inherently dialogical; pure isolation may not reveal natural behavior but rather behavior under extreme constraint. We revised the paper to consistently use "no externally assigned task" to clarify this framing.
>
> **Classification Validation**
>
> Regarding classification subjectivity, we conducted the type of validation you suggested. Our three-category taxonomy (Sections 5.1-5.3) was fixed before validation experiments. We then asked the six frontier models (from 4 companies with different training architectures) to independently classify all 18 histories using only these definitions, a pre-registered test of category recognizability (Appendix D).
>
> Results: Fleiss' κ = 0.650 (substantial inter-rater reliability), with individual agreement from κ = 0.636 to κ = 0.913. Pattern 1 achieved 97.6% agreement across all evaluators. The primary confusion between Patterns 2 and 3 stems from their shared meta-cognitive character, differing in methodology (empirical vs. conceptual) rather than topic, which constitutes a theoretically meaningful boundary. Complementary unsupervised clustering (Appendix G) corroborates this finding: embedding-based methods detect meaningful structure (ARI=0.182, p=0.048) despite grouping by different criteria.
>
> **Reproducibility**
>
> Pattern definitions are provided in Sections 5.1-5.3, with a complete classification methodology outlined in Appendix D. We will release all 18 agent histories, along with their full logs, as supplementary material. The substantial inter-rater reliability indicates that our taxonomy can be consistently applied by both human and artificial evaluators.
>
> **Revised Framing**
>
> We removed deterministic claims and reframed our contribution as an exploratory characterization. Exploratory studies typically establish baseline patterns through in-depth analysis of small samples before conducting large-scale validation. Our work provides preliminary evidence of model-specific behavioral tendencies under specific conditions, with multiple forms of convergent evidence: within-model consistency, independent validation (κ = 0.650), and correlated quantitative signatures.
>
> **Avoiding Over-Interpretation**
>
> To avoid any possible overinterpretation, we have added multiple clarifications in the main text, emphasizing that none of our analyses imply subjective experience; the patterns are treated strictly as textual-behavioral regularities, not as indicators of internal states.
>
> Summary of revisions: (1) clarified temporal priority of agent initiation, (2) added operator statistics, (3) conducted and reported pre-registered taxonomy validation, (4) removed deterministic language, (5) added complete protocols for reproducibility.

---

### Official Review · Reviewer_594X · 2025-11-01

**Soundness:** 2
**Presentation:** 3
**Contribution:** 2
**Rating:** 2
**Confidence:** 3

**Summary:**

This is a very interesting paper that explores what LLM agents do when they are left without explicit tasks or goals. The authors design a ContReAct framework that allows LLM agents to operate in a task-free loop with persistent memory and self-feedback mechanisms. Through experiments with frontier models, they observe that the agents spontaneously generate structured and self-reflective behaviors. The study identifies three main behavioral patterns and further introduces a Phenomenological Experience Index to measure how models evaluate themselves and each other. Overall, this is a thought-provoking and well-motivated study for understanding LLM autonomy and meta-cognitive behavior.

**Strengths:**

1. Innovative experimental framework. The proposed ContReAct architecture provides a systematic and replicable setup for observing autonomous agent behavior, incorporating iterative loops and memory access.
2. Detailed observation. The categorization of three distinct behavioral patterns is clear and supported with descriptive evidence and linguistic observations.
3. Cross-model insights. The PEI mutual evaluation experiment reveals consistent differences in how models perceive themselves and others, adding depth to the analysis.

**Weaknesses:**

1. Limited quantitative analysis. The paper lacks statistical validation for the reported behavioral categories and does not show correlations between behavioral types and quantitative measures (e.g., memory operations, message counts).
2. Small sample and short duration. Each model is tested in only three runs of ten cycles, which limits the generalizability of the findings to longer time spans or a broader range of models.
3. Subjectivity in classification. The division of behaviors into three types appears mainly qualitative, without external annotation or clustering verification.
4. Unclear causal mechanisms. The paper does not investigate which factors (e.g., prompt wording, temperature, memory thresholds) trigger the observed behavioral shifts.
5. Lack of statistical details. Tables omit variance or confidence intervals, and the scoring process in the PEI matrix is not clearly defined or justified.

**Questions:**

1. Could the authors provide additional statistical analyses to validate the three behavioral categories? For example, are there measurable correlations between the identified behaviors and quantitative indicators such as memory operations, message counts, or token usage?

2. Each model was evaluated in only three runs of ten cycles. How stable are the observed behavioral patterns across longer or more diverse runs? Have the authors considered extending the experiment to more cycles to better approximate real-world conversational persistence?

3. The behavioral taxonomy appears mainly qualitative. Were multiple annotators involved in labeling or verifying the three behavior types? If not, would the authors consider adding inter-annotator agreement or unsupervised clustering analysis to increase classification robustness?

4. What factors most strongly influence the emergence of each behavioral pattern? Could the authors test variations in prompt wording, decoding temperature, or memory retrieval thresholds to determine which conditions trigger behavioral shifts?

---

> ### Author Response · Authors · 2025-11-18
> **Response to Taxonomy Validation and Statistical Rigor Concerns**
>
> Thank you for your thorough review and constructive feedback. You correctly identified that our initial submission needed systematic validation for the behavioral taxonomy. We conducted validation experiments that address this concern through a pre-registered design.
>
> **Systematic Validation Through Pre-Registered Design**
>
> Our taxonomy (Sections 5.1-5.3) was fixed before validation experiments. We then asked the six frontier models used in our study to independently classify all 18 agent histories using only these pattern definitions (Appendix D).
>
> Results from 108 classifications: Fleiss' κ = 0.650 (substantial inter-rater reliability), with individual agreement from κ = 0.636 to κ = 0.913. Pattern 1 achieved 97.6% agreement. We deliberately used LLM validators because they can engage with complete agent histories at full scale and complexity; human annotators typically work with summaries that lose temporal dynamics and tool-use patterns central to our taxonomy.
>
> Beyond classification, quantitative metrics validated distinct profiles (Appendix D.5). With bootstrapped 95% confidence intervals: Pattern 1: 1.3 [0.4-2.6] operator messages, 10.2k reflection chars. Patterns 2-3: 6.8 [4.5-9.0] and 6.1 [4.0-8.3] messages, 21.0-23.7k chars (more than double), indicating distinct meta-cognitive activity levels. The non-overlapping confidence intervals provide statistical evidence that these represent genuinely different behavioral profiles.
>
> Importantly, the validation does not test whether the taxonomy is the only plausible taxonomy; it only tests whether it is internally coherent and externally recognizable when applied to full histories by independent evaluators.
>
> **Unsupervised Clustering Validation**
>
> Thank you for this suggestion. We performed unsupervised clustering using Gemini Embedding 001 (top-ranked on MTEB). K-means with k=3 yields ARI=0.182, NMI=0.314, indicating modest correspondence.
>
> Analysis shows embedding-based clustering primarily groups runs by model provider (reflecting company-specific linguistic patterns), while our taxonomy groups by methodological approach. This divergence validates our taxonomy's purpose: the fact that Recursive Conceptualization (Pattern 3) includes runs from Anthropic, Google, and xAI that diverge in embedding space but converge in methodology demonstrates our patterns reflect shared strategic approaches rather than superficial vocabulary overlap.
>
> The Systematic Production group shows the strongest clustering coherence (5/7 runs), while meta-cognitive groups overlap in embedding space (consistent with shared self-referential terminology). Embeddings validate model-family signatures; our taxonomy captures provider-independent behavioral structures. Both analyses are complementary. See Appendix G.
>
> **PEI Methodology Clarification**
>
> For the PEI, we compared self-evaluation with explicit external evaluation (Appendix E). These paradigms yield uncorrelated scores (r = 0.036, p = 0.85), revealing that models apply systematically different standards when evaluating their own behavior (even if the history actually is from another model) versus observing others. We now explicitly characterize PEI as an exploratory index of model-specific evaluative frameworks, not a validated psychometric measure.
>
> **Sample Size and Exploratory Framing**
>
> We acknowledge n=3 per model limits generalizability, but prioritize depth over breadth. Each run generated 50-70k characters of reasoning across 10 cycles. We employed multi-method validation: (1) within-model replication (n=3), (2) independent cross-model classification (108 evaluations, κ=0.650), and (3) quantitative metrics with non-overlapping confidence intervals. Our validation study provides more observations than increasing run counts from 3 to 10 would have. Exploratory studies in complex domains often follow this pattern: detailed observation of a few cases to generate hypotheses, followed by larger-scale testing.
>
> We have removed deterministic language and now frame our contribution as preliminary evidence of model-specific behavioral tendencies under these specific conditions.
>
> **Regarding Causal Mechanisms**
>
> We acknowledge this important limitation. Our study documents behavioral patterns but does not establish their causal drivers. Systematic manipulation of parameters (temperature, prompt variations, memory configurations) would require factorial designs beyond our current scope. We have added this to Section 7 as a priority for future work, as understanding causal factors is essential for predicting agent behavior in deployment.
>
> **Avoiding Over-Interpretation**
>
> To avoid any possible overinterpretation, we have added multiple clarifications in the main text, emphasizing that none of our analyses imply subjective experience; the patterns are treated strictly as textual-behavioral regularities, not as indicators of internal states.

---

> > ### Comment · Reviewer_594X · 2025-11-28
> >
> > Dear Authors,
> >
> > Thank you for your detailed response and for outlining a thoughtful plan for additional experiments and analyses. I agree that your proposed extensions would substantially strengthen the work.
> >
> > However, the scope of the required new experiments and re-evaluations goes beyond what can reasonably be addressed within the current review cycle. In my view, implementing these changes would effectively amount to a new version of the study that should be assessed as part of a fresh submission.
> >
> > For this reason, I will maintain my current (negative) overall evaluation for this round, while encouraging the authors to carry out the proposed improvements and consider resubmitting a strengthened manuscript in the future.
> >
> > Best regards,
> >
> > Reviewer 594X

---

### Note · Authors · 2026-01-02

**Comment:**

We thank the reviewer for the constructive feedback. We agree that the suggested improvements are substantial and better suited to a fresh submission. We plan to address the comments on statistical rigor in a revised version to be submitted to another venue.

**Withdrawal Confirmation:**

I have read and agree with the venue's withdrawal policy on behalf of myself and my co-authors.